# Paediatric Post-Traumatic Stress Risk for Young People and Parents following Acute Admission for Paediatric Multisystem Inflammatory Syndrome: Retrospective Analysis of Psychological Screening and Support

**DOI:** 10.3390/children11070858

**Published:** 2024-07-15

**Authors:** Anita Freeman, Emily Golding, Jennifer Gardner, Zoe Berger

**Affiliations:** 1Great Ormond Street Hospital, NHS Foundation Trust, London WC1N 3JH, UK; emily.golding@gosh.nhs.uk (E.G.); zoe.berger@stgeorges.nhs.uk (Z.B.); 2Leeds Community Healthcare NHS Trust, Leeds LS11 0DL, UK; jennifer.gardner11@nhs.net

**Keywords:** post-traumatic stress, paediatric intensive care, PIMS-TS, medical traumatic stress

## Abstract

Background: Paediatric Multisystem Inflammatory Syndrome (PIMS-TS) is a novel condition that was identified for the first time during the COVID-19 pandemic. Limited research exists that describes the psychological needs of children and young people (CYP) following an acute hospital admission for PIMS-TS. Methods: A retrospective cohort study was conducted to explore both psychological need and access to psychology services for CYP and their families who were admitted to a paediatric tertiary hospital in London, UK, for PIMS-TS between April 2020 and May 2021. Results: We included 121 CYP and a parent/caregiver for each. In total, 23.3% of the CYP were at risk of developing a traumatic stress response and 11.6% were at risk of experiencing emotional disturbance. Of the parents screened, 40.5% also scored above clinical cut-off for a trauma response. There was a significant relationship observed between CYP and parents identified as having a trauma risk. Conclusions: The importance of proactive screening for both trauma and emotional distress in CYP and their parents/caregivers following acute hospital admission is highlighted. In addition, there is a need to think about a CYP as part of a system of care and to ensure that clinicians pay attention to parental wellbeing and mental health when understanding the psychological impact on a child.

## 1. Introduction

For most children and young people (CYP), a COVID-19 infection causes a mild respiratory illness [1] with low rates of hospital admission and a low risk of mortality [2]. However, in April 2020, a novel inflammatory condition in CYP was identified, termed Paediatric Multisystem Inflammatory Syndrome (PIMS-TS).

The first case definition was released by the UK Royal College of Paediatrics and Child Health (RCPCH) in May 2020, defining PIM-TS as a “persistent fever, inflammation and evidence of single or multi-organ dysfunction in a child, with exclusion of any other microbial cause with or without PCR evidence of SARS-CoV-2”. It is a rare condition, with estimates suggesting that it occurs in less than 0.5% of children who have (or who had) COVID-19 [3]. Despite this, in the first wave of the pandemic, affected children required hospitalisation and most of them required Paediatric Intensive Care (PICU) [4]. The data suggests that young people from different social and ethnic backgrounds were differentially affected, with children in global majority groups, living in more deprived areas and in key worker families being over-represented [5].

There has been research into the clinical features, causes, treatment and management of CYP with a PIMS-TS diagnosis [3,4,6]. However, to date, little has been published on the psychological effects of PIMS-TS for CYP and their families. 

London-based tertiary children’s hospitals include a paediatric psychologist in their PIMS-TS MDT as part of routine care [5,7], initially as inpatient support and later following discharge. Clinical observations have identified a number of common themes amongst young people affected. These included feelings of isolation, uncertainty about the trajectory and recovery from illness and shame and stigma associated with the condition [7] 

The limited research conducted to date has highlighted psychological distress and trauma symptoms following discharge from hospital, at 6–8 weeks [7] and at 6 months [8]. This evolving picture regarding the psychological needs of CYP with PIMS-TS is not surprising given all children in the early waves of the pandemic experienced an intensive care admission [4,9]. PTSD diagnosis and symptomology has been shown to be significantly more common in children who have been admitted to an intensive care unit compared to general wards [10,11]. Research has also shown that the psychological sequalae is still present 1 year following admission [12] and risk factors for ongoing distress have been found to be associated with illness severity, clinical scores at 3 months post-admission and parental anxiety [13]. There is emerging evidence highlighting the needs of the parent(s) of children admitted to PICU with research identifying that a significant minority (between 21% and 32% of parents meeting cut off on screening) of parents experience post-traumatic stress symptoms following the PICU admission of their child [14,15,16] and that children’s trauma symptoms are positively correlated with those of their parents [10,11,13]. The research findings suggest that parents experiencing post-traumatic stress disorder is “related to parents’ perceptions of the threat to their child’s life and to acute stress reactions to the PICU” [14,17,18,19]. 

The importance of psychological monitoring for parents and CYP following PICU admission is widely acknowledged [13]. There is also a recognition that trauma in the form of PTSD or PTSS (post-traumatic stress symptoms) can follow devastating diagnoses, such as childhood cancer [20]. It has also been recognised that there are considerable psychological sequalae that are the result of physical health diagnoses in CYP and their families. The main purpose of this current study is to provide an insight into the psychological needs of CYP with this novel syndrome and their parents after discharge from hospital. It was hypothesised that as all had a PICU admission, a significant number of the CYP would report PTSD symptoms and emotional distress. Some of the results of this study have been reported previously in the form of an abstract/presentation [7]. It is important to note that screening was carried out during a period of heighted global anxiety in the context of national lockdown during the COVID-19 pandemic. At this point, little was known or documented about the possible long-term impact of the pandemic on CYP’s mental health more generally. 

The current research has implications for health care resources and the care provided. It is hoped that a better understanding of the psychological needs of this population will inform psychological care and improve the outcomes of children with a significant health condition and their parents/care givers [21].

## 2. Materials and Methods 

### 2.1. Participants

This was a retrospective cohort study of psychological screening and referral to psychological services for patients with PIMS-TS and their families. This study includes all patients aged 18 years or younger meeting diagnostic criteria for PIMS-TS [22] who were admitted a paediatric tertiary hospital in London, UK, between April 2020 and May 2021. 

### 2.2. Procedure

A full ethics review under the terms of the Government Arrangements of Research Ethics Committees in the UK was not required as the data analysis was retrospective, and no additional data was collected beyond that collected for standard medical care. A service evaluation was registered with the NHS Trust Audit Committee (registration number 3223). Demographics and medical data were obtained from the CYP’s medical record. 

The 121 CYP included in this study would have typically become unwell at home before presenting to their local hospital without having had any prior knowledge of PIMS-TS before being transferred to a specialist centre with a Paediatric Intensive Care Unit (PICU). This admission will have taken place between April 2020 and May 2021 during a period of national COVID-19 pandemic restrictions. The patient pathway is illustrated in Figure 1 below. The typical PICU stay was 2–4 days before being moved to a paediatric ward for on-going care, typically for a further 5–7 days (admission average 9.34 days and ranged from 2 to 28 days). During their hospital admission, both CYP and their parent/carer would have had access to the paediatric psychology service through a referral from their nursing or medical team. After discharge home, the CYP would be invited to follow-up outpatient PIMS-TS Multi-Disciplinary Team (MDT) review appointments at 2 weeks, 6 weeks, 6 months and 1 year post-discharge. Paediatric psychology was an integrated part of the PIMS-TS MDT and all parents/carers would have been made aware that they could access separate psychological appointments at any point during their CYP’s care.

All families were given a brief psychological wellbeing screening as part of their 6-week outpatient PIMS-TS MDT review appointment. All parents and CYP consented at the point of being given the measures. Screening measures included those for post-traumatic stress symptoms and emotional distress for the CYP (aged 8 years and over). In addition, one parent or carer (referred to as parent throughout the text for clarity) for CYP of all ages was asked to complete a post-traumatic stress screening measure. Further psychological follow-up support was offered to all CYP and/or parents who scored above the screening cut off, and/or who identified any psychological distress during the psychology review.

A retrospective case note review of all psychology inpatient and outpatient referrals for PIMS-TS for CYP and parents was completed to provide a summary of psychological support offered to these families, and the number of subsequent follow-up psychology appointments was calculated. This included any documented reason or identified need related to the psychological support.

### 2.3. Measures

The screening questionnaires used during the psychology review were selected based on brevity, as well as their validity for use with a paediatric population that had been admitted to PICU after sudden onset of symptoms. The purpose of the measures was to identify any persisting psychological distress after the CYP returned home and was no longer acutely unwell. The screening measures administered are detailed below:Children’s Revised Impact of Events Scale (CRIES-13): Children’s post-traumatic stress symptoms were measured using the 13-item version of the CRIES. This is a self-report measure for use with children aged 8 years and above, with established reliability and validity (α = 0.80) [23]. A total score of 30 means that 75–83% of children with a PTSD diagnosis have been correctly identified [23]Paediatric Index of Emotional Distress (PI-ED): Emotional distress, including anxiety and depression symptoms, was measured using the PI-ED screening measure (O’Connor, et al., 2010). This is a 14-item self-report measure for children aged 8 to 18 years, with established reliability and validity in paediatric population (α = 0.86) [24]. A total score of 20 or above indicates a CYP experiencing clinically significant levels of distress [25].Impact of Events Scale-Revised (IES-R): Post-traumatic stress symptoms were screened for in parents using the IES-R. This is a self-report measure used for adults aged 18 years and above, with established reliability and validity (α = 0.96) [26]. A cut-off score of 24 or more was used as an indication of ‘clinical concern’ for post-traumatic stress symptoms [27].

### 2.4. Data Analysis

Descriptive statistics were used to describe the patient demographic and admission information, psychological screening scores and number of psychology sessions attended following a PIMS-TS hospital admission.

Non-parametric statistics were used to analyse the data. This was due to not all variables being normally distributed. A chi-square test of independence was used to identify any relationship between parents’ and children’s post-traumatic stress scores, between CYP trauma and emotional distress screening scores, and between ethnicity grouping and referral to psychology. Analyses were performed using the SPSS version 24 software (SPSS, Inc., Chicago, IL, USA).

## 3. Results 

### 3.1. Demographics

A total of 121 CYP were admitted to the hospital to receive acute care for PIMS-TS between April 2020 and May 2021. The average age was 9.3 (median 9, interquartile range 5). Of the CYP, 70 were recorded as male and 51 as female. Figure 2 below shows the breakdown of the CYP population by ethnicity recorded in patient records. The majority of those admitted were identified in patient records as being from Black African, Afro-Caribbean, Asian or other UK minority ethnic backgrounds (N82, 67.8%), which will be classified as Global Majority Ethnicity Backgrounds from here on. We identified 21 (17.4%) of the CYP as being from White or European ethnic backgrounds and for 18 CYP the ethnicity was documented as ‘Other’ or was not recorded (14.9%). All CYP lived in the city of London or surrounding North London regions.

### 3.2. Hospital Journey

The hospital admission ranged from 2 to 28 days, with a median of 9.3 days. This included both the intensive care and the subsequent ward admission. The outpatient PIMS-TS MDT reviews took place at 2 weeks, 6 weeks, 6 months and 12 months post-discharge, as described by [8]. A psychological review and completion of screening measures took place at the 6-week outpatient appointment.

### 3.3. Psychological Screening

A retrospective descriptive analysis of the psychology review screening measures was completed to provide a summary of the CYP’s (aged 8–17 years) and their parent’s psychological needs at 6–8 weeks post-discharge. Most of the screenings took place at a 6-week PIMS-TS MDT review appointment, with an average length post-admission of 7 weeks (SD 2.8, range 2–20 days).

Of the 121 CYP, 86 were aged 8 years or older, and eligible to complete CRIES-13 and PI-ED screening measures. A total of 60 CYP (70%) completed the CRIES-13 and the PI-ED, and a total of 79 (65%) of 121 parents of CYP also completed the IES-R screening questionnaire.

In total, 40.5% of the parents and 23.3% of the CYP met the cut-off value on the traumatic stress screening (IES-R and CRIES-13) and 11.6% of CYP met the cut-off value on the measure of emotional distress (PI-ED) (Table 1). Those who did not complete the measures either did not consent to screening or were not available to meet the psychologist during their MDT review. 

A chi-square test of independence was performed to examine the relationship between the CYP meeting CRIES-13 cut-off and the parent meeting IES-R cut-off. The relationship between these CYP and parent trauma screenings was significant, χ^2^ (1, N = 50) = 12.346, *p* = 0.0001, suggesting that a relationship exists between child and parent post-traumatic stress risk.

The number of CYP meeting the cut-off for emotional distress on the PI-ED was lower (11.6%) than those reaching the cut-off for PTSD on the CRIES-13 (23.3%). The relationship between the CYP screening measure cut-offs was assessed using a chi-square test of independence to examine the relationship between CYP aged 8 years and older meeting CRIES-13 trauma cut-off and meeting the PIED emotional distress cut-off. The relationship between these variables was not significant, χ^2^ (1, N = 59) = 4.058, *p* = 0.044, suggesting these screening measures identify different aspects of psychological distress, which supports the potential value of using both measures for screening this population of CYP.

### 3.4. Referrals to Psychology

A retrospective case notes review of psychology inpatient and outpatient referrals and subsequent number of psychology sessions recorded in medical records was completed. This showed that a total of 64 (53%) of the 121 CYP included in this study were referred for psychological support either during admission or post-discharge with an average of 5.10 (SD 4.53) psychology sessions (telephone/video/in-person) and a range of 1–20 sessions.

Of the 64 CYP referred to psychology, 58 had complete documentation regarding the reason for referral and psychology support offered, which is summarised below in Table 2 and Table 3. Six of the CYP’s records were excluded due to incomplete or inconsistent documentation.

The most frequent reasons for referral to psychology were emotional distress, anxiety and making sense of hospital experience (Table 2). Trauma was mentioned in only 7% of referrals, despite the screening showing 40.5% of parents and 23.30% of the CYP met cut-off for clinical indication of PTSD (Table 1).

Of the 58 reviewed referrals, the majority (89.6%) included a request for the CYP to access psychology support, and only 8.6% of referrals were a request for parent/carer support only (Table 3).

Of the 58 referrals reviewed, 31% were categorised as assessment, liaison or consultation only (1–2 sessions), 43.1% as brief interventions (2–6 psychology sessions) and 25.8% as extended psychology intervention (7–20 sessions) (Table 4).

Due to the variability in the format and recording practices of psychology sessions, a systematic qualitative review of the themes that emerged in the psychology sessions was not possible. However, it was noted that the psychology interventions offered were often in relation to the themes of anxiety and emotional distress in the context of managing the uncertainty and fear for the future with a new and unknown diagnosis within a pandemic context, and to a lesser extent the potential for isolation, stigma and shame and not feeling able to share or talk about their PIMS-TS experience with friends and family.

The likelihood of a CYP being referred to psychology was broken down by ethnicity grouping, as illustrated in Figure 3 below. This shows a lower percentage of psychology referrals made for CYP identified as having White British and White Other ethnic (33%) backgrounds compared to the other ethnicity groupings (Asian, 70.6%; Black African and Afro-Caribbean 42.1%; other ethnic minority groups 56.3%).

A chi-square test of independence was performed to examine the relationship between ethnicity grouping (Global Majority Ethnicity Backgrounds or White UK and Other) and referral to psychology. The relationship between ethnicity and whether a CYP/family were referred to psychology was significant, χ^2^ (1, N = 109) = 10.288, *p* = 0.001. Those CYP identified as being from Global Majority Ethnicity Backgrounds were more likely to be referred to psychology than those classified as being from White ethnic backgrounds.

## 4. Discussion

This study shows that approximately 23.3% of CYP presenting to a tertiary paediatric hospital during the early stages of the COVID-19 pandemic with a novel disease called PIMS-TS were at risk of developing a traumatic stress response and 11.6% were at risk of experiencing emotional disturbance. A total of 40.5% of the parents screened also scored above the clinical cut-off value for a trauma response. All CYP had an admission to a PICU. The psychological impact found in this current study is in line with research looking at the emotional sequelae of CYP following PICU admissions [11], and of those with other paediatric conditions [20]). The data also showed positive significant correlations between the trauma scores of a CYP and their parent. Again, this finding is well documented in the paediatric literature [20,21].

Important clinical themes for CYP that have emerged from this current data include increased feelings of uncertainty about the illness and the illness trajectory as well as feelings of isolation, fear for future and potential shame and stigma.

A total of 55.4% of the CYP were referred to the psychology service either during or after their admission to hospital and received an average of five sessions with a psychologist. The reason for referral to psychology from MDT colleagues rarely mentioned trauma as a presenting concern, yet a significant proportion of the CYP were experiencing trauma-related symptoms. In addition, only 8.6% of the referrals were for psychological support for the parent, yet the data suggested a significant proportion were experiencing distress.

When exploring the relationship between the referrals to psychology and the demographic of the population, the data indicates that a significantly greater proportion of CYP from a Global Majority ethnic background were referred to the service. It is unclear why this difference was observed. It could be hypothesised that this is related to the identified referral themes of isolation, stigma and shame acting as increased psychological risk factors for those from these backgrounds compared to those from UK and other White backgrounds.

There are several limitations to the data, including that this was a single centre audit of a tertiary children’s hospital based in London. There is a possibility of referral bias, with referral rates observed to be higher for the most unwell patients with PIMS-TS [8] and those from particularly disadvantaged backgrounds due to regional demographics. Despite these limitations, there are several important clinical implications from this retrospective review. It once again highlights the importance of proactive screening for both trauma and emotional distress in CYP and their parents/caregivers following sudden and unexpected hospital admission, especially to intensive care units. It also highlights the need to think about a CYP as part of a system of care and to ensure that clinicians pay attention to parental wellbeing and mental health when understanding the psychological impact on a child.

## 5. Conclusions

These results shine a spotlight on the potentially important role medical teams have in containing and ameliorating unnecessary distress in CYP and their families following a physical health diagnosis. At the point at which the CYP in this sample were being admitted with this novel disease and during the height of the pandemic, there was limited knowledge about the trajectory, prognosis or potential recovery from the illness. The medical team were, therefore, unable to provide necessary information and/or reassurance to CYP and their families both to navigate the current admission or to plan for the future. Preventative interventions which might otherwise have been available in the form of leaflets, psychoeducation and psychologically informed resources were not yet available. This understandably might have accounted for higher levels of anxiety and distress in this population. In addition, there were a number of CYP and parents who reported feelings of isolation, fear and uncertainty as well as shame and stigma. This related to feeling as if they could have avoided contracting the illness, that they would be perceived to be ‘to blame’ for becoming unwell and/or had concerns about contamination and how they would be experienced by others once discharged. Clinically, this felt reminiscent of the clinical themes that emerge when working with CYP with an HIV diagnosis [28]. We recognise that there were no opportunities for CYP and their parents to mix or meet with other families on the ward when admitted, whereby peer-to-peer support would usually be fostered. Lastly, we think it is important to note that a large proportion of the CYP with PIMS-TS were from a UK Global Majority background. We wondered if this meant that a number of the CYP had already had experiences of discrimination, marginalisation and structural racism that heightened their experience of shame and stigma. As already discussed in a previous paper [29], there were considerable benefits from the CYP being able to meet with other CYP who had PIMS-TS and talk about their experiences. This had positive benefits on wellbeing by reducing isolation and helping CYP to connect with stories of recovery and growth.

## Figures and Tables

**Figure 1 children-11-00858-f001:**
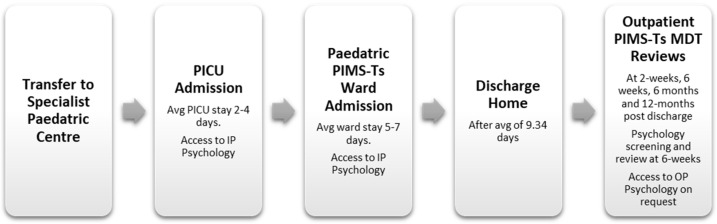
Typical patient pathways for PIMS-TS paediatric patients.

**Figure 2 children-11-00858-f002:**
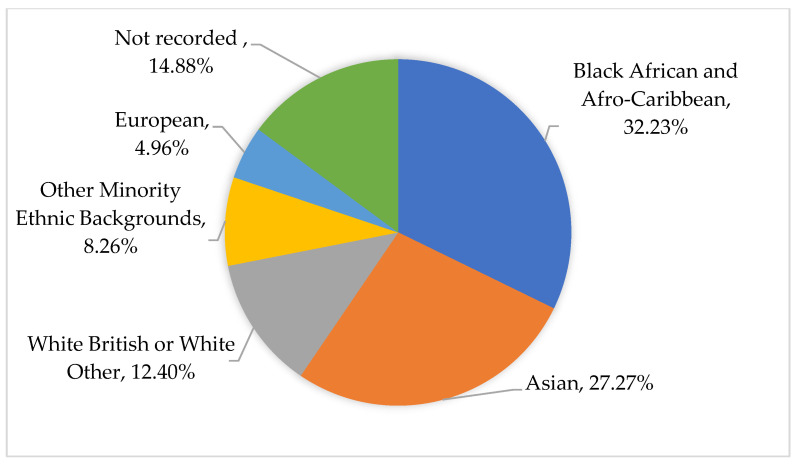
Ethnicity of CYP as self-identified and recorded in patient records.

**Figure 3 children-11-00858-f003:**
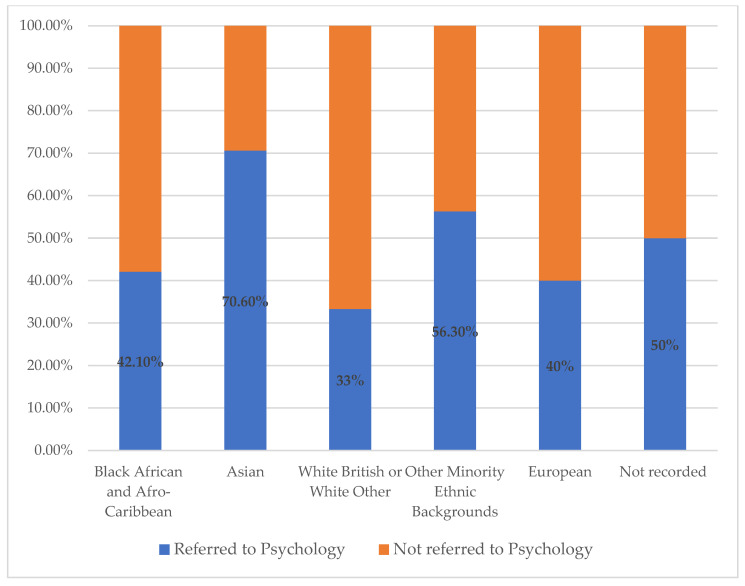
Percentage of PIMS-TS patient population referred to psychology by Global Majority ethnic background and White/European categories.

**Table 1 children-11-00858-t001:** Number and percentage of those screened meeting clinical cut-off on post-discharge psychology screening measures.

Screening Measure	N	% of Those Screened
IES-R (Parent measure)	32	40.5%
CRIES-13 (CYP measure)	14	23.3%
PI-ED (CYP measure)	10	11.6%

**Table 2 children-11-00858-t002:** Number and percentage of referrals by referral reason, as documented in patient records.

Referral Reason	N	%
Emotional distress	10	17.2%
Anxiety	10	17.2%
Processing/making sense of hospital experience	10	17.2%
Low mood	5	8.6%
Trauma	4	6.9%
Withdrawal/isolation	3	5.2%
Behavioural difficulties	3	5.2%
Adjustment to new diagnosis	3	5.2%
Psychology review request	3	5.2%
Managing/Coping with condition	3	5.2%
Cognitive/Neurodevelopmental concerns	1	1.7%
Extended psychology assessment	1	1.7%
Procedural Anxiety	1	1.7%
Appearance related concerns	1	1.7%
**Total**	**58**	**100.0%**

**Table 3 children-11-00858-t003:** Number and percentage of psychology referrals by who the request was for.

Referral Request for	N	%
Patient/CYP	30	51.7%
Patient and Parent/Carer	22	37.9%
Parents/Carers	5	8.6%
Not specified	1	1.7%
**Total**	**58**	**100.0%**

**Table 4 children-11-00858-t004:** Length of psychology intervention by number of psychology sessions documented in patient records.

Length of Psychology Intervention	%	N	Average No. Psychology Sessions
1–2 sessions (assessment/liaison/consultation only)	31.0%	18	1.33
3–6 sessions (brief intervention)	43.1%	25	3.96
7–20 sessions (extended intervention)	25.8%	15	12.07
**Total**	**100.0%**	**58**	**5.10**

## Data Availability

The data presented in this study are available on request from the corresponding author. The data are not publicly available due to privacy reasons.

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
