# Peer review of "Paediatric Post-Traumatic Stress Risk for Young People and Parents following Acute Admission for Paediatric Multisystem Inflammatory Syndrome: Retrospective Analysis of Psychological Screening and Support"

_children, 2024, doi:10.3390/children11070858_

Round 1

Reviewer 1 Report

Comments and Suggestions for Authors

Thank you for sharing your manuscript. I think your research is very interesting and useful, but I have some concerns about how you describe the results of this research. Please see my comments:

1.               There is no description of the participants in the abstract. Please indicate the total number of children and parents who participated in the study.

2.               Although you are using a chi-square test, the sample is quite small and specific. It would be good to describe the preliminary calculations of sample size and statistical power.

3.               Please specify Cronbach's alpha coefficients for your study.

4.               The section "Materials and methods" is difficult to perceive without the appropriate subsections ("Participants", "Measures", "Data analysis").

5.               When describing the measures, I would more clearly identify those that are intended for children and the instrument for parents.

6.               For some reason, data analysis is presented in the Results. Please transfer this information to the Materials and methods.

7.               Table 1 is unclear. If you do not read the text, then it is not clear where the children filled in, where the parents are. It would be nice to sign this in the table.

8.               Please note: you have a larger font in the tables than in the main text.

9.               In Figure 3, it would be good to indicate the percentages. Without them, the figure looks uninformative.

10.            I recommend moving Clinical implications from Conclusion to Discussion, prescribing them after the limitations of the study.

I hope that my comments will help you improve your manuscript.

Sincerely yours,

reviewer.

Author Response

Thank you for reviewing out manuscript.  we have jointly discussed your comments and responded as below.  

  1. There is no description of the participants in the abstract. Please indicate the total number of children and parents who participated in the study.

Thank you for highlighting this. We have added this information into the abstract and reformatted the abstract – Line 9-24.

  1. Although you are using a chi-square test, the sample is quite small and specific. It would be good to describe the preliminary calculations of sample size and statistical power.

Thank you for this suggestion. We chose a chi-square under statistical advise to look at the relationships between the different screening measurement cut offs. We recognise that the sample is small and specific, but this was a retrospective study, so a power calculation was not conducted. We sought statistical advice and were advised at the time of deciding on analysis possible and decided to conduct a chi-square.

  1. Please specify Cronbach's alpha coefficients for your study.

Thank you, we have added these as suggested to the measures section. Line 133-150

  1. The section "Materials and methods" is difficult to perceive without the appropriate subsections ("Participants", "Measures", "Data analysis"). 

Thank you, we have added these subheadings to help structure for the reader. See line 85-162

  1. When describing the measures, I would more clearly identify those that are intended for children and the instrument for parents.

Thank you for this comment, we have noted the measures are clearly identified as ‘child’ measures or ‘parent / adult’ measures in the Psychological Screening Measures section. See line 135-150 and in table 1 – see line 197  

  1. For some reason, data analysis is presented in the Results. Please transfer this information to the Materials and methods.

Thank you, we have moved this to the methods section. See line 152-162

  1. Table 1 is unclear. If you do not read the text, then it is not clear where the children filled in, where the parents are. It would be nice to sign this in the table.

Thank you, we have amended this.

  1. Please note: you have a larger font in the tables than in the main text.

Thank you, we have re-formatted the font size on all tables – See line 196-237.

  1. In Figure 3, it would be good to indicate the percentages. Without them, the figure looks uninformative.

Thank you, please note that the % are shown on the y-axis. We have added further clarification into the text to help with this, and added further data labels to the graph to further illustrate. See line 244-253

  1. I recommend moving Clinical implications from Conclusion to Discussion, prescribing them after the limitations of the study.

Thank you, we have moved this. See line 289-199

Reviewer 2 Report

Comments and Suggestions for Authors

The paper focuses on PTSD symptoms and psychological interventions on pediatric patients with PIMS-TS and their parents. This is an important topic not so present in the literature.

I have only some minor concerns.

The introduction is limited. Could you add more studies? If the studies are limited you can add PTSD symptoms in children with other pediatric diagnosis.

It is also important to stress more the COVID period as traumatic one that could develop different type of negative symptoms.

An informed consent should be use for parents in screening psychological symptoms. Was it given for the study? Or for the clinical assistance?

The instruments should be addressed more, including validity and reliability.

What are the reasons for drop-out of your participants? What is the response rate?

The relationship between CYP aged 8 years and older CRIES-13 trauma cut-off and the PIED emotional distress cut-off is significant if the p value is 0.04. Is it right? If it is so, you have to reconsider it in your results and discussion.

The discussion of the results is limited and could be more ample, including possible preventing interventions and an explanation on why the trauma problems are underestimated.

Author Response

Thank you for reviewing and providing feedback on our paper.  we have all carefully read your comments and responded as below and in the revised manuscript.  We hope you approve of the changes.

The introduction is limited. Could you add more studies? If the studies are limited you can add PTSD symptoms in children with other pediatric diagnosis. It is also important to stress more the COVID period as traumatic one that could develop different type of negative symptoms.

Thank you, we have added references for broader pediatric diagnoses associated with trauma.

We have further noted the comment suggesting further mention of the COVID context. See line 65-84. 

An informed consent should be use for parents in screening psychological symptoms. Was it given for the study? Or for the clinical assistance?

We are not sure what is meant by ‘clinical assistance.’ To clarify this is a retrospective review and all children, young people and parents consented and engaged in the psychological screening (all measures included) as part of the child’s routine MDT pediatric care – we have clarified this in the Methods. See line 112-120

The instruments should be addressed more, including validity and reliability.

Thank you, we have added in Cronbach's alpha coefficients to the measure descriptions.  Please see line 121-151. 

What are the reasons for drop-out of your participants? What is the response rate?

Thank you for your questions.  We have specified the n and % of people who completed the measures (of those who were eligible) – please see line 189-192.

We did not have ‘drop-outs’ as this was a retrospective review of patient routine care. But the dat included people who did not consent to screening or were not available or able to complete the screening in their MDT review appointment due to clinical reasons. We have added further clarification for this – see line 194-196.

The relationship between CYP aged 8 years and older CRIES-13 trauma cut-off and the PIED emotional distress cut-off is significant if the p value is 0.04. Is it right? If it is so, you have to reconsider it in your results and discussion.

Thank you for querying this cut off. We chose to use a 0.01 significance in SPSS, which gave a non-significant output. We chose to be cautious with level of significance given the small sample size. We are familiar with both CRIES and PIED measures, which have been developed and validated to measure different constructs of pediatric distress with PIED and trauma for CRIES. We would therefore conclude that 0.01 significance was most appropriate in this instance.

The discussion of the results is limited and could be more ample, including possible preventing interventions and an explanation on why the trauma problems are underestimated

We had not wanted to go too far into recommendations beyond what the data is telling us. However, we have added further discussion about the development of resources to help with this. Please see lines 289-326